# Resistance to Chemotherapeutic 5-Fluorouracil Conferred by Modulation of Heterochromatic Integrity through Ino80 Function in Fission Yeast

**DOI:** 10.3390/ijms241310687

**Published:** 2023-06-26

**Authors:** Kim Kiat Lim, Nathaniel Zhi Hao Koh, Yi Bing Zeng, Jun Kai Chuan, Raechell Raechell, Ee Sin Chen

**Affiliations:** 1Department of Biochemistry, Yong Loo Lin School of Medicine, National University of Singapore, Singapore 117596, Singapore; bchlimk@nus.edu.sg (K.K.L.); e0412949@u.nus.edu (N.Z.H.K.); yibingzzz@u.nus.edu (Y.B.Z.); junkai@u.nus.edu (J.K.C.); bchrae@nus.edu.sg (R.R.); 2National University Health System (NUHS), Singapore 119228, Singapore; 3NUS Center for Cancer Research, Yong Loo Lin School of Medicine, National University of Singapore, Singapore 117599, Singapore; 4NUS Graduate School-Integrative Sciences & Engineering Programme, National University of Singapore, Singapore 119077, Singapore

**Keywords:** 5-FU, chemotherapeutic drug, resistance, heterochromatin, fission yeast, *Schizosaccharomyces pombe*

## Abstract

5-Fluorouracil (5-FU) is a conventional chemotherapeutic drug widely used in clinics worldwide, but development of resistance that compromises responsiveness remains a major hurdle to its efficacy. The mechanism underlying 5-FU resistance is conventionally attributed to the disruption of nucleotide synthesis, even though research has implicated other pathways such as RNA processing and chromatin dysregulation. Aiming to clarify resistance mechanisms of 5-FU, we tested the response of a collection of fission yeast (*Schizosaccharomyces pombe*) null mutants, which confer multiple environmental factor responsiveness (MER). Our screen identified disruption of membrane transport, chromosome segregation and mitochondrial oxidative phosphorylation to increase cellular susceptibility towards 5-FU. Conversely, we revealed several null mutants of Ino80 complex factors exhibited resistance to 5-FU. Furthermore, attenuation of Ino80 function via deleting several subunit genes reversed loss of chromosome-segregation fidelity in 5-FU in the loss-of-function mutant of the Argonaute protein, which regulates RNA interference (RNAi)-dependent maintenance of pericentromeric heterochromatin. Our study thus uncovered a critical role played by chromatin remodeling Ino80 complex factors in 5-FU resistance, which may constitute a possible target to modulate in reversing 5-FU resistance.

## 1. Introduction

Unregulated proliferation of malignant cells is a classic hallmark of cancer. In this regard, the ability to sustain DNA synthesis, and DNA repair mechanisms in the presence of replication stress, is vital to cell viability and proliferation. This process is largely dependent on a sufficient balance of deoxyribonucleotides (dNTPs) upon initiation of replication in the S phase of the cell cycle [1]. A loss of this balance can lead to a mis-incorporation of nucleotides into replicating DNA, while a depletion of dNTPs can lead to replication fork stalling, both of which are detrimental to genomic stability [2]. Failure to resolve these issues could eventually lead to cell death and in metazoan, p53-mediated apoptosis [3]. 

Tight regulation of dNTP level balance in the cell is essential for proper progress of DNA replication and repair of DNA damage [1]. Timely regulation of synthesis of various dNTPs is integral to DNA synthesis and replication for cell cycle progression. One of these, deoxythymidine triphosphate (dTTP), is synthesized via conversion of deoxyuridine monophosphate by thymidylate synthase to deoxythymidine monophosphate, which is then phosphorylated to deoxythymidine triphosphate by a cascade of kinases [4]. 

Owing to the central role played by dNTP synthesis, this pathway constitutes a key chemotherapeutic target for anti-cancer therapy. One of these classic chemotherapeutic inhibitor agents is the fluorinated pyrimidine analogue 5-Fluorouracil (5-FU). Since its initial synthesis in 1957 [5], it has been used in the treatment of a variety of cancers, and is especially efficacious against solid tumours including head and neck squamous cell carcinomas and colorectal cancer [6,7]. After entering the cells, 5-FU is converted into three main active metabolites: fluorouridine triphosphate (FUTP), fluorodeoxyuridine triphosphate (FdUTP) or fluorodeoxyuridine monophosphate (FdUMP) [8]. FUTP and FdUTP can mis-incorporate into RNA and DNA, respectively, thus interfering with RNA transcription, DNA synthesis and repair. On the other hand, FdUMP irreversibly binds and inhibits thymidylate synthase [9]. These converge to DNA-replication disruption in the S-phase of a cell cycle and DNA damage [10]. The thymine insufficiency leads to nucleotide imbalance from a build-up of toxic dUTP, which eventually can result in a “thymineless” cell death [11,12].

At the cellular level, disruption of DNA and RNA sequences arising from 5-FU metabolites’ mis-incorporation can lead to broad-ranging effects that have not yet been fully characterized [13]. For example, 5-FU mis-incorporation into RNA can interfere with the catalytic Rrp6 subunit of exosome, thereby reducing the ribonucleolytic activity of the complex and compromising RNA turnover [14]. This effect may go beyond inhibiting the processing of rRNA, tRNA and mRNA, as it was shown in yeast to affect long non-coding RNAs and antisense RNA [15] and it is suggested to include also non-coding RNA derived from silenced heterochromatic repeats [16]. 

Constitutive heterochromatin is formed to compact and silence repetitive sequences in order to maintain genomic stability. However, the non-coding transcripts arising from the transcription of these repetitive sequences by RNA polymerase II (RNAPII) is essential for establishing the silenced heterochromatin on these loci, and such regulation is especially well studied in fission yeast [17,18,19,20,21]. The RNAPII-derived transcripts are converted into double stranded RNAs by RNA-dependent RNA polymerase [22], which in turn serve as the substrate for Dicer to generate small interference RNA (siRNA) [23,24,25]. The siRNA associates with Ago1 protein and incorporates into the RNA-induced transcriptional silencing (RITS) complex via the Ago1-siRNA chaperone (ARC) complex [24,26,27]. The RITS-associated single-stranded siRNA base pairs with the RNA transcription arising from the heterochromatic repeats thus directs the RITS to these heterochromatic regions [24,26,27]. The Clr4 methyltransferase complex (Clr-C) is subsequently recruited by RITS via physical interaction to methylate the histone H3 lysine 9 (H3K9) to constitute the binding site for chromodomain (CD) motif-containing proteins Swi6 and Chp2. These CD proteins form a platform to recruit more silencing complexes such as the Clr3-containing Snf2/Hdac-containing repressor complex (SHREC) to maintain the silenced heterochromatin [28,29,30]. 

In the attempt to understand the genetic determinants underlying 5-FU responsiveness, we studied the influence of 5-FU on the growth of previously identified null mutants of multi-environmental factor responsiveness (MER) genes in fission yeast. We previously reported mutants of these 91 MER genes exhibit hypersensitivity to various DNA-damaging agents and environmental factors including cations [31,32,33,34,35]. Because our approach involved a manual serial dilution spot test, we decided to begin the screen on this smaller collection of MER gene mutants as these strains have been reliably checked and backcrossed to generate a prototrophic genetic background [31]. From our screen, we observed mutants of genes encoding chromosome segregation, DNA repair and membrane transporters, and biosynthesis enzymes of coenzyme Q10, which function in mitochondrial oxidative phosphorylation to be required for cellular tolerance against 5-FU. Many of these mutants were also identified in previous reported screens [16]. However, we surprisingly observed mutants for the subunits of the chromatin remodeler Ino80 complex to be resistant to 5-FU. Additionally, the disruption of the Ino80 factors remedied the deficiency in the mutant RNA-interference (RNAi) factor Ago1 in 5-FU tolerance and upheld chromosome-segregation fidelity. We further showed that this phenotypic complementation occurred at the chromatin level through restoration of silencing of the heterochromatic DNA sequences. This novel observation suggests the possibility that chromatin architectural modulation at the heterochromatic pericentromere is a plausible target to modulate 5-FU susceptibility.

## 2. Results

### 2.1. Cell Growth Assay of MER Gene Mutants to Identify Genes Involved in 5-FU Sensitivity and Resistance

To broaden the understanding of how cells resist the cytotoxic effects of 5-FU, we studied the hypersensitivity response of previously reported MER gene mutants towards 5-FU. Serial dilution spot test assay was performed to compare the growth of the MER gene mutant strains against the WT strain. Asynchronous log-phase cultures of each strain were 10-fold serially diluted and spotted onto YEA plates of various concentrations of 5-FU: 50, 100, 150 μM. DMSO solvent was added to control plates to serve as a non-drug control. This range of drug concentrations was chosen based on an initial screen to capture the range effective in visualizing both sensitivity and resistance responses to the drug. Cell growth was examined at day 3 and day 7, which corresponded to the intermediate and saturated stage of growth as previously reported [31,34] (Figure 1A, Appendix A). 

The growth fitness of the 91 MER gene mutants on media containing 5-FU was computed into a sensitivity score (S-score) as previously done [34], which took account of the dose-response growth on various drug concentrations relative to that of untreated subjects, and further normalized to the WT of similar genetic background. A positive score (bars pointing to right, Figure 1A) indicates that the loss of that gene results in resistance to 5-FU, while a negative score (bars pointing to left, Figure 1A) reflects sensitivity. A score close to zero implies a growth response similar to the WT strain (Figure 1A). In total, 12 mutants were identified as resistant, 5 were identified as severely sensitive, 7 moderately sensitive, 36 mildly sensitive and lastly 31 were identified as non-sensitive/resistant (Figure 1, Appendix A). Strains that showed a 100-fold sensitivity or greater across multiple drug concentrations were indicated as severely sensitive while sensitivity less than 10-fold were rated mild, and moderate sensitivity was between these two ranges. Strains that displayed improved growth over the WT strain on 5-FU drug plates at either time point were classified as resistant. The spotting assay for these MER gene mutants on 5-FU media was repeated at least three times to ensure reproducibility of the phenotypes. 

The most sensitive genes belong to the ontology group chromatin remodeling, transcription-related and chromosome segregation (Figure 1, Appendix A), consistent with what was observed in a previous whole-genome screen that reported on 5-FU perturbing fidelity of chromosome partitioning in the cell cycle [16]. Of note, our study identified mutants of the DASH complex (*Δdad1*, *Δdad2*, *Δdad3*, *Δdad5*, *Δduo1*, *Δspc19*) [36,37] to be mildly to moderately sensitive to 5-FU (Figure 1, Appendix A). As expected from the interference of 5-FU on DNA replication, mutants of all DNA repair factors (*Δctp1*, *Δmhf1*, *Δmhf2*, *Δmms1*, *Δrad32*, *Δrhp51*, *Δrhp54*, *Δssb3*, *Δrad24*) showed hypersensitivity to 5-FU, even though their low sensitivity was rather unexpected. This group included mutants of homologous recombination repair genes (*Δrhp51*, *Δrhp54*), and that encode factors involved in stalled replication fork repair (*Δmhf1*, *Δmhf2*, *Δrad32*). These mutants, in conjunction with those of genes involved in deoxyribonucleotide (dNTP) synthesis (*Δcsn1*, *Δcsn2*, *Δcdt2*), supported the mis-incorporation of 5-FU to create mismatching DNA helices that results in disrupted replication progress, which is the proposed major mode of action of 5-FU in fission yeast, as in humans [38,39]. Besides these expected classes of hypersensitive mutants, our study also detected mutants of genes encoding membrane-bound transporters (*Δerd2*, *Δnpp106*, *Δrav1*, *Δspcc18.02*, *Δvph2*, *Δvps35*, *Δvps901*), mitochondria-related, especially the coenzyme Q10 (CoQ10) biosynthesis enzymes (*Δcoq2*, *Δcoq3*, *Δcoq4*, *Δcoq6*, *Δcoq7*, *Δdps1*) [40], and translation regulators (*Δdph2*, *Δsce3*, *Δspac6g9.14*, *Δspbc19g7.10c*) to be hypersensitive to 5-FU (Figure 1A, Appendix A). Twenty-four of the 5-FU hypersensitive mutants reported here were also identified in the screen previously performed by Mojardin et al. (2015): *Δrpa12*, *Δnht1*, *Δiec1*, *Δada2*, *Δngg1*, *Δgcn5*, *Δcph2*, *Δies6*, *Δrsc1*, *Δrsc4*, *Δarp42*, *Δrhp51*, *Δmhf1*, *Δduo1*, *Δspc19*, *Δmhf2*, *Δdad1*, *Δdad3*, *Δcor1*, *Δapl5*, *Δcsn1*, *Δspbc19g7.10c*, *Δcox19*, and *Δppr1*. 

There were mutants of 12 genes demonstrating resistance to 5-FU with individuals identified from several ontological classes. Interestingly, more than half (58.3%, 7 out of 12) of these genes encode subunits of the chromatin-remodeling complex Ino80 family genes (*iec1^+^*, *iec3^+^*, *ies2^+^*, *ies4^+^*, *ies6^+^*, *nht1^+^*, *hap2^+^*/*SPCC16C4.20c*). The remaining 5-FU-resistant genes regulate pathways in dNTP synthesis (*ada1^+^*, *ccr4^+^*, *SPAC2F3.11*) and transcription (*nrm1^+^*, *yox1^+^*), whereas *pmd1^+^* encodes a multidrug resistance membrane transporter. 

As this has not been noticed in previous studies focusing on a large cluster of genes surrounding 5-FU resistance, the Ino80 family of genes were selected for further study.

### 2.2. Mutants Involved in Centromere Stability Are Sensitive to 5-FU

Several genome-scale chemogenomic analyses have been performed to look at 5-FU responsiveness in fission yeast [16,41,42,43], but none of these null mutants of Ino80 complex subunit genes were observed to exhibit resistance against the drug. Ino80 complex, an ATP-dependent nucleosome-remodeling complex, was shown recently to regulate the integrity of constitutive heterochromatin, which encompassed the centromere of the fission yeast to confer stable kinetochore attachment to spindle microtubules for proper chromosome segregation during the M-phase of the cell cycle [44]. Changes have been observed on pericentromeric heterochromatin when cells are treated with 5-FU and growth of mutants of several heterochromatin-associating factors was compromised in 5-FU [42]. This previous study showed some mutants of the heterochromatin-binding factors to be more affected by 5-FU. We therefore assess a broad panel of mutants to see if there may be some sub-pathways that when compromised result in higher 5-FU sensitivity. To this end, we ten-fold serially diluted *Δrdp1*, *Δago1*, *Δtas3*, *Δcid12*, *Δclr4*, *Δrik1*, *Δraf2*, *Δstc1*, *Δclr3* and *clr6-1*, which are mutants of factors functioning in pathways including RNA interference (RNAi) (*RITS: ago1*, *tas3; RDRC: rdp1*, *cid12*) [24,45], histone H3 lysine 9 methylation (*clr4*, *rik1*, *raf2*) [30], histone deacetylation (*clr6-1*, *clr3*) [28,46], on 0, 50–150 µM 5-FU (Appendix A). Overall, the majority of the mutants tested were hypersensitive to 5-FU with the exception of *clr6-1*. Among these, the RNAi mutants showed higher hypersensitivity, especially *Δago1* which exhibited approximately 1000-fold less growth on 50 µM 5-FU relative to untreated control (Appendix A). Histone deacetylase mutants appeared to be rather irresponsive to 5-FU with *Δclr3* showing 10-fold sensitivity only at 150 µM 5-FU, while *clr6-1* remained resistant (Appendix A). 

### 2.3. Mutants of Certain Ino80 Gene Family Members Partially Suppress 5-FU Sensitivity in Heterochromatin Mutants

The loss of heterochromatin structure in cells lacking Dicer that forms short interference RNA to enable RNAi-mediated heterochromatin assembly was reported to result in enhancement of histone exchange enabled by the Ino80 complex [44]. Since 5-FU treatment led to heterochromatic changes correlated with chromosome-segregation defects [16], we subsequently checked whether the genetic changes that result in 5-FU resistance may also impact the RNAi and H3K9me pathways. To this end, we combined the 12 null mutations that conferred 5-FU resistance (*Δyox1*, *Δnrm1*, *Δspac2f3.11*, *Δada1*, *Δpmd1*, *Δccr4*, *Δnht1*, *Δhap2*, *Δiec1*, *Δies2*, *Δies4* and *Δiec3*) with *Δago1* and *Δclr4* and compared the growth of the resultant double mutants to the parental single mutants using serial dilution spotting assay on 5-FU-containing media. 

All the null mutants were able to suppress the 5-FU hypersensitivity of *Δago1*, except *Δnrm1* and *Δyox1* (Figure 2, Appendix A). Among these mutants, Δ*nht1* and Δ*iec1* showed the strongest suppression, which was already apparent at the intermediate growth stage (Day 3). In contrast to the mutants of these ‘accessory’ Ino80 subunit genes [47], that of other subunits, including an *ies2* ‘core’ subunit, exhibited lower suppression of *Δago1* on 5-FU, slightly at day 3 but only apparent at day 7 on 5-FU plates (Figure 2, Appendix A). Other 5-FU-resistant mutations also suppressed *Δago1*, yet to a much lesser extent than the Ino80 subunit mutants (Figure 2, Appendix A). 

We also tested whether the 5-FU-resistant mutations can suppress *Δclr4*. Genetic interaction with *Δclr4* mutants were more varied. Among the mutants of the Ino80 factors, only Δ*ies4* resulted in suppression of the Δ*clr4* strain on 5-FU, while mutants of the other subunit genes of Ino80, namely Δ*nht1*, Δ*hap2*, Δ*iec1*, and Δ*ies2*, exhibited cumulative 5-FU hypersensitivity when Δ*clr4* was concomitantly deleted, whereas Δ*iec3* did not show genetic interaction (neither suppressing nor synthetic lethal) (Figure 3, Appendix A).

With regards to resistant mutants other than the Ino80 subunit genes, the genetic interaction with *Δclr4* background was similarly varied. While *Δyox1* and *Δpmd1* did not show genetic interaction, Δ*nrm1* was synthetic lethal with *Δclr4* on exposure to 5-FU. However, of note, *Δspac2f3.11* was able to suppress *Δclr4* 5-FU hypersensitivity phenotype, as *Δago1* (Figure 3, Appendix A).

### 2.4. 5-FU-Resistant Mutations Also Suppressed TBZ Sensitivity of Δago1 

RNAi mutants host disruption of centromeric heterochromatin that compromises kinetochore function, resulting in hypersensitivity to microtubule depolymerizing drug thiabendazole (TBZ) [48,49]. To assess if the suppression of *Δago1* and *Δclr4* may similarly result in the TBZ hypersensitivity of these mutants, we performed serial dilution spotting assays on 8 and 15 µg/mL TBZ, to check TBZ sensitivity of the single and double mutants of 5-FU-resistant gene mutants combined with *Δago1* and *Δclr4*. The β-tubulin mutant *nda3-KM311* was employed as the positive control [49]. 

The TBZ hypersensitivity of the double mutants mirrored the sensitivity phenotype of the genetic interaction with *Δago1* on 5-FU (Appendix A). The only notable difference was that *Δccr4* that was able to suppress *Δago1* on 5-FU showed no similar suppression on TBZ. In the double mutants with *Δclr4*, sensitivity phenotypes were once again varied but bore similarities with that of 5-FU (Appendix A). While *Δiec3* and *Δies2* were able to suppress TBZ sensitivity of *Δclr4* (as on 5-FU), no genetic interaction was observed for *Δnht1* with *Δclr4* (Appendix A). Overall, it seemed that the 5-FU-resistant mutants showed similar genetic relationship with *Δago1* and *Δclr4* on both 5-FU and TBZ, suggesting that the suppression of these mutants may be mediated via the complementation of the chromosome-segregation defects. 

### 2.5. Loss of Ino80 Function Promotes Chromosome-Segregation Fidelity of Δago1

To examine whether the loss of Ino80 function may indeed complement the chromosome-segregation defect in 5-FU, we conducted fluorescence microscopy to compare the nuclear segregation phenotype between *Δago1* in the presence and absence of the genes encoding two subunits of Ino80 complex, *iec1^+^* and *ies4^+^*. The strengths of suppression of these mutants were among the strongest and one (*iec1^+^*) expresses a fission yeast-specific subunit while *ies4^+^* encodes a conserved subunit [47,50]. 

Log phase cells of WT, *Δago1*, *Δiec1*, *Δies4*, *Δago1Δiec1* and *Δago1Δies4* strains were treated with 500 μM 5-FU for 4 h, with cells treated with DMSO serving as a control to ensure this solvent did not contribute toxicity that interferes with chromosome segregation. Nuclear staining was achieved with DAPI and cells with normal mitotic phenotypes as well as that with chromosome missegregation were counted. The nuclear phenotypes were also assessed in cells with single *Δyox1*, *Δnrm1* and *Δiec3* mutations and the respective double mutants in which *Δago1* was concomitantly deleted. *Δyox1* and *Δnrm1* mutants did not suppress *Δago1* (Figure 2) and hence were chosen as negative controls. Among these, *Δiec3* is a mutant of another yeast-specific Ino80 subunit [47]. *Δago1* hosted rather pleiotropic improper chromosome segregation, including cells with lagging chromosomes, which is associated with many heterochromatin-defective mutants being observed (Figure 4A) [51]. 

In the untreated asynchronous *Δago1* mutant cell culture, approximately 34.7% of mitotic and post-mitotic cells (cells hosting >1 nucleus) exhibited missegregation of chromosome that phenotypically deviates from WT cells with equally sized segregated nuclei (Figure 4A,B, *Δago1*, black bar). This proportion of cells showing abnormal chromosome partition was increased to 51.6% when *Δago1* cells were cultured for 4 h in 500 µM 5-FU (Figure 4B, *Δago1*, white bar). Chromosome missegregation was also elevated in *Δyox1* and *Δnrm1* but the frequency was unchanged in 5-FU (Figure 4B). The double mutant derived from deleting *ago1^+^* in *Δyox1* and *Δnrm1* (*Δyox1Δago1* and *Δnrm1Δago1*, respectively) showed a level of chromosome missegregation comparable to *Δago1* single mutant, which was also not further increased upon exposure to 5-FU (Figure 4B). In contrast, deletion of genes encoding the Ino80 complex subunits Iec1 and Ies4 resulted in significant near two-fold reduction of chromosome segregation aberration from 51.6 ± 8.7% of *Δago1* to 35.9 ± 6.2% and 35.9 ± 4.1% in *Δago1Δiec1* and *Δago1Δies4,* respectively (Figure 4C, white bars). Unlike *iec1* and *ies4*, the mutant of another Ino80 factor *iec3* did not suppress the chromosome missegregation defects of *Δago1* upon disruption (Figure 4D). 

### 2.6. Ino80 Disruption Restores Heterochromatic Silencing Defect of Δago1 at Pericentromere

Stable capturing of spindle microtubule onto kinetochore to promote precise chromosome segregation during mitosis relies on the stability of centromeric chromatin [18,52,53]. Centromere in WT fission yeast cells consists of long stretches of constitutively silenced heterochromatin and chromatin status at such pericentromeric heterochromatic loci was shown to undergo visible changes in cells treated with 5-FU [16]. Due to the importance of Ago1 in ensuring transcriptional silencing of pericentromeric heterochromatin, we proceeded to assess whether the compromise of Ino80 that suppressed the chromosome-segregation defects of *Δago1* may be associated with the restoration of transcriptional silencing at the outer centromeric repeat sequence. 

Employing reverse transcription PCR, we checked the transcript status at the *dh* repeat (*otr*) compared to the constitutively transcribed *act1* gene locus in the euchromatin [21,54]. A low level of transcript was detected from the *dh* repeat sequence in the untreated WT cells that was upregulated more than 3-fold in *Δago1*, but was partially restored (reduced 59.5%) when *iec1* was concomitantly deleted. Similar suppression was not evident in the case of *ies4*. Interestingly, we observed that pericentromeric *dh* repeat become more silenced when genes encoding these Ino80 subunits were deleted (Figure 5A). The trend of suppression, however, was more readily observed under the +5-FU condition. The *Δago1* mutant exhibited much augmented derepression of the pericentromeric heterochromatic region that was significantly suppressed in both *Δago1Δiec1* and *Δago1Δies4* mutants (Figure 5B). Taken together, these results showed that deletion of *iec1* and *ies4* can enforce integrity of pericentromeric heterochromatin. This restoration of the transcriptional silencing of the repetitive sequences therein very likely contributes to the suppression of chromosome-missegregation aberrations in *Δago1* cells. 

## 3. Discussion

This work aims to contribute to the understanding of 5-FU mechanisms of action by identifying genes affecting susceptibility of cells towards the drug using fission yeast as a model. Such insights have the possibility of generating hypotheses for managing drug resistance and cytotoxicity that may be applicable to human cells via uncovering yet unknown target pathways of this chemotherapeutic agent, or re-sensitization upon the acquisition of resistance. Drug responsiveness at the molecular level poses a complex problem in which studies such as that performed herein may provide valuable insights.

### 3.1. Susceptibility to 5-FU Results from Disruption to dNTP Metabolism, Mitochondrial Functions and Membrane Transport Genes

Our screen identified hypersensitivity in mutants hosting deletions of genes regulating dNTP metabolism, mitochondrial function and intracellular membrane-bound transporters. Obtaining genes regulating dNTP metabolism is consistent with the integrity of dNTP synthesis towards the tolerance of 5-FU cytotoxicity [30]. Ample and balanced supply of dNTP is essential for synthesis and repair of DNA. Csn1, Csn2 and Cdt2 are components of Pcu4-Ddb1-COP9 signalosome (CSN) reported to activate ribonucleotide reductase (RNR) through the nuclear-to-cytoplasmic translocation of the small subunit of RNR via the degradation of RNR-inhibiting Spd1 factor. Cells that do not express *csn1*, *csn2* and *cdt2* were observed to stabilize Spd1 and RNR inhibition [55,56,57], which may underlie the 5-FU sensitivity of these mutants. It was reported that knockdown of CSN6 subunit of human CSN can confer chemosensitivity through enhancing the defects of nucleotide synthesis of colorectal cancer cells [58]. 

Our results concurred with that of a previously reported screen to identify null mutant cells of several intracellular transporters to show 5-FU hypersensitivity [59]. We also isolated *Δvps35* and *Δerd2* showing 5-FU sensitivity (Figure 1) [59]. Vps35 forms part of the retromer complex that functions in endosomal protein transport from endosome to Golgi apparatus and plasma membrane that leads to retrieval and recycling of transmembrane receptors and other proteins between these compartments [60,61], whereas Erd2 is a transmembrane receptor that acts in endoplasmic reticulum (ER)-retention of luminal proteins that typically contain a KDEL or HDEL motif. Erd2 functions in retrieval of these ER luminal proteins from the Golgi apparatus [62,63]. In silico structural modeling predicts Erd2 to serve as a cargo receptor in protein trafficking [64]. It is not clear why intra-cellular transport factors such as retromers and K(H)DEL receptors are involved in 5-FU response. This could be mediated via proteins that are sorted through pathways regulated by these receptors, or retention of proteins within the ER lumen or removal of such proteins from Golgi may be required for the drug sensitivity. However, in support of a link of the intracellular protein traffickers and 5-FU response, a report has shown that the adaptor that links the retromer to its regulator, FAM21, physically associates with several factors in the NFkB pathway and its depletion sensitized pancreatic cancer cells to the 5-FU via disruption of the chromatin association of the NFkB transcription factor [65]. 

The previous study isolated mutants of AP-1 adaptor subunits—*Δapl2* and *Δapl4*—to be 5-FU sensitive [59]; we did not, however, observe similar sensitivity for the two AP-1 receptor mutants (*Δapl5* and *Δapl6*) used in our study. On the other hand, we identified mutants of the Rav1 subunit of the RAVE complex and Vph2, which are assemblers of vacuolar ATPase (V-ATPase) [32,34] to show high sensitivity to 5-FU. V-ATPase is a proton pump that mediates acidification of sub-cellular compartment subunits that is also present on the ER membrane [66,67]. The *Δrav1* and *Δvph2* have been found to also show hypersensitivity to other cytotoxic agents in fission yeast including doxorubicin [31,68], which are sensitive to pH in the cytosol [69]. Expression and activation of V-ATPase is correlated with chemoresistance and inhibition of V-ATPase has been suggested to be a means to mediate re-sensitization of tumor cells to chemotherapeutic agents [70,71,72]. 

Our previous study revealed V-ATPase synergizes with the multidrug resistance transporter p-glycoprotein (Pmd1 in fission yeast) to confer resistance to doxorubicin in fission yeast and human cells [32]. Unexpectedly, *Δpmd1* exhibited resistance towards 5-FU instead of sensitivity like towards doxorubicin. While the reason for this phenomenon is unclarified, drawing from an in silico prediction with doxorubicin versus epirubicin that differed by the presence of a hydroxyl side chain to confer differential P-gp interaction, it is possible that the structure of 5-FU may be unfavorable for interaction with Pmd1 [32]. 

### 3.2. The Roles of Chromatin Organization and Chromosomal Segregation in 5-FU Sensitivity

Similar to a previous study [16], we isolated numerous mutants of genes encoding regulators of chromosome segregation and organization to be sensitive to 5-FU. The overlap included *gcn5* and *ngg1* that encode histone acetyltransferase (HAT) complex subunits and *arp42* that encodes an actin-related protein associated with both SWI/SNF and RSC complexes. Loss of function of these chromatin-remodeling complexes leads to vastly differential expression profiles as these complexes regulate both transcriptional activation and repression [73]. Another protein common with the Mojardin screen is *yaf9*, a mutant of SWR1 histone H2A.Z-chaperoning chromatin remodeler and NuA4 HAT. These complexes often modulate chromatin organization at the promoter to regulate transcription [74]. It stands to reason that disruption of these genes could alter gene expression that could lead to repression or enhancement of important pathways essential for 5-FU susceptibility, which may include cellular processes such as DNA repair, metabolism and organelle development [74]. 

Interestingly, we also showed that mutants of all subunits of the DASH complex included within our screen were hypersensitive to 5-FU. The DASH complex is an oligomer composed of 8–10 subunits that form a ring structure to facilitate attachment of microtubules to kinetochores for proper bi-orientated attachment of sister chromatids to mitotic spindles during chromosomal segregation [57]. Interference of replication was shown in budding yeast to result in DNA defects compromising microtubules with respect to kinetochore attachment in tubulin mutants [75]. It is possible that 5-FU interferes with DNA replication and could likewise synergize with mutation of the DASH proteins to disrupt mitotic segregation integrity. 

### 3.3. Loss of Ino80 Function Conferred 5-FU Resistance by Counteracting Loss of Heterochromatin Integrity

Proper segregation of chromosomes also relies on the integrity of the pericentromeric heterochromatin, which underlies the microtubule-capturing kinetochore complex and withstands the exertional force of the spindle microtubules on the centromere [76]. One novel group of 5-FU-resistant genes identified in our screen is the over-represented Ino80 complex subunit genes. Deletion mutants of these genes resulted in resistance to 5-FU. Our analysis discovered that loss of integrity of pericentromeric heterochromatin, such as in the *Δago1* strain, resulted in hypersensitivity to 5-FU. However, deleting genes encoding various subunits of the Ino80 complex can interestingly overcome 5-FU susceptibility, correlating with the restoration of heterochromatic integrity as observed by the partial restoration of transcription repression at the pericentromeric heterochromatic *dh* repeat (Figure 5) accompanied also by a partial suppression of chromosome missegregation frequency (Figure 4). 

The Ino80 complex has been shown to promote histone turnover that is important to evict nucleosomes at promoter and DNA damage sites to enable binding of transcription machinery and DNA repair factors [77,78]. Recently, Ino80-associated histone turnover activity has also been linked to the formation and epigenetic inheritance of specialized chromatin, which include the CENP-A chromatin of inner centromere and heterochromatin at pericentromere [44,79]. Mutants of the accessory subunits of Ino80 but not the conserved subunits alleviated the silencing deficiency in a cell without Dicer protein in TBZ [44]. We also observed a similar trend here whereby the mutant of the accessory subunit Iec1 (*Δiec1*) suppressed *Δago1*, while the mutant of the conserved subunit Ies4 (*Δies4*) did not (Figure 5A). However, both mutants were observed to suppress the defects of *Δago1* in the presence of 5-FU (Figure 5B), suggesting a functional remodeling within the Ino80 complex that may be induced by the presence of the drug. Nevertheless, *Δago1* defect was still preferentially complemented by the absence of accessory subunits more than the conserved subunit observable on hypersensitivity to the microtubule-destabilizing agent TBZ (Appendix A). Interestingly, we observed partial suppression of TBZ hypersensitivity exhibited by a mutant strain of the H3K9 histone methyltransferase (*Δclr4*) upon deletion of some subunits of Ino80 (*Δies2*, *Δiec3* and *Δies4*), but not for the high-mobility group (HMG) domain-hosting subunit (*Δnht1*) [33]. 

It is possible that the suppression of histone turnover may retard the loss of histones from the pericentromeric heterochromatin, which may occur during DNA damage [78] and in doing so preserve the H3K9 methylated histones essential for maintaining the integrity of the transcriptionally repressed chromatin to ensure proper chromosome segregation under the 5-FU challenge. Our work thus raise the interest of developing pharmacological inhibitors to overcome 5-FU resistance that commonly occurs with prolonged use of this effective anti-cancer agent. 

## 4. Materials and Methods 

### 4.1. Fission Yeast Manipulation 

Standard procedure of fission yeast manipulation was followed [80,81]. Cell growth was carried out in YEA media (3% glucose, 0.5% yeast extract, 75 mg/L adenine) (Sigma-Aldrich, St Louis, MO, USA). Solid media plates contained 2% Bacto-agar (BD Bioscience, Franklin Lakes, NJ, USA). Genetic crosses were performed by mixing haploid cells of opposite mating types on SPA plates and meiotic progenies dissected using automated dissection microscope MSM400 (Singer Instruments, Somerset, UK) as previously described [21,82,83]. MER strains were obtained from previous screens that examined sensitivity to multiple cytotoxic agents, chemotherapeutic drugs and cation [31,33,34,35]. These are haploid prototrophic strains and arise from backcrossing commercially purchased single-gene knockout library strains (Bioneer, Daejeon, Republic of Korea) with prototrophic WT strains. Gene disruption was confirmed via PCR using gene-specific primers as described previously [31]. All strains were stored as a frozen stock in glycerol at −80 °C.

### 4.2. Serial Dilution Spotting Cell Growth Assay 

Previously published procedure and analysis approaches was followed [34]. In brief, yeast strains revived from frozen stock were grown on YEA plates, and then inoculated into YEA media with shaking aeration overnight at 30 °C. Asynchronous-log phase cells were diluted in liquid YEA to an optical density of 0.5 (OD_600_) to standardize the concentration of cells, and subsequently 10-fold serially diluted and spotted onto plates with or without drugs. Growth was documented at intermediate (day 3) and stationary (day 7) growth stages upon incubation at 26 °C for temperature-sensitive strains or 30 °C otherwise. Stock solutions of 5-FU and TBZ were made by dissolving in DMSO (Sigma-Aldrich), and either used fresh or aliquoted to store at −20 °C with a maximum of one freeze–thaw cycle to minimize degradation. An appropriate volume was added to YEA media containing desired concentrations of the drugs. The drug sensitivity of the strains was quantified using sensitivity score (s-score), which describes the drug dosage response normalized to that of WT as previously reported [34,35].

### 4.3. Fluorescence Microscopy 

Standard microscopy procedure was carried out as previously described [82,84,85,86]. Log-phase cells were treated with the appropriate concentrations of drug for 4 h at 30 °C before fixation in 1/10 volume glutaraldehyde on ice for 15 min, washed three times with cold 1x phosphate-buffered saline (PBS). Cells were then mixed with equal volume of 50 µg/mL 4′,6-diamidino-2-phenylindole (DAPI) (Thermo Fisher Scientific, Waltham, MA, USA) and nuclear phenotypes visualized using Plan-Apo 100× objective lens on a Nikon Ti-E inverted microscope (Nikon, Tokyo, Japan). Images were taken using Nikon NIS-Elements AR software (v5.02).

### 4.4. Statistical Analysis 

Statistical significance was assessed via Student’s *t*-test and denoted by *p* values smaller than 0.05 using Microsoft Office Excel 365 software. 

### 4.5. Reverse Transcription Polymerase Chain Reaction (RT-PCR) 

TRIzol (Thermo Fisher Scientific, Waltham, MA, USA) was used to extract total RNA from log-phase growing cells. The RNA preparation was treated with DNase I (New England Biolabs, Ipswich, MA, USA), which was then removed via phenol:chloroform:isoamyl alcohol (25:24:1) (Nacalai Tesque, Kyoto, Japan) extraction. A 100 ng amount of DNase I-treated RNA was used for one-step RT-PCR with OneStep RT-PCR kit (Qiagen, Venol, The Netherlands), as previously described [54,82]. qPCR was subsequently done with iTaq Universal SYBR Green Supermix (Bio-Rad Laboratories, Hercules, CA, USA) using the StepOne real-time PCR system (Thermo Fisher Scientific).

## Figures and Tables

**Figure 1 ijms-24-10687-f001:**
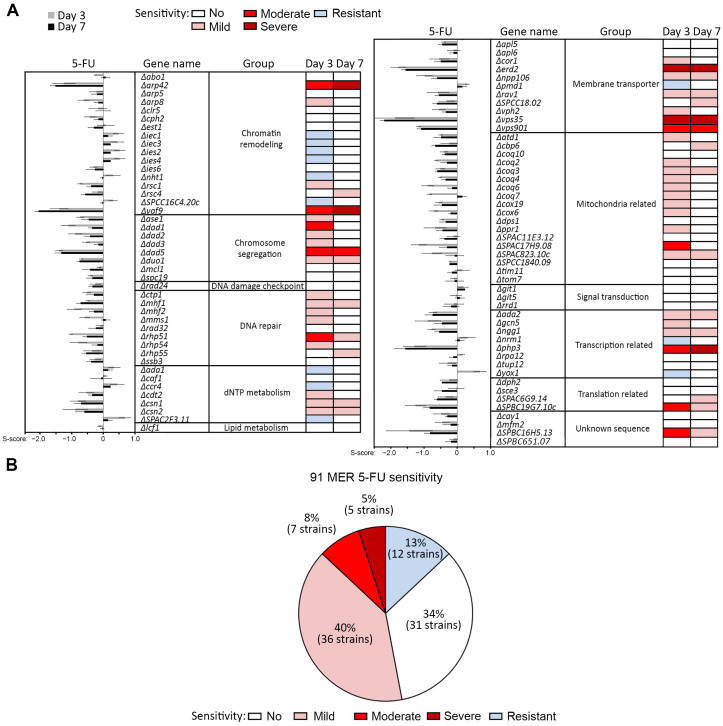
Consolidation of the sensitivity score (s-score) results depicting dose-response effect of the MER gene mutants to 5-FU that were documented on days 3 and 7 after drug exposure. (**A**) Bar graphs denote s-score values while color codes represent different levels of drug sensitivity: dark red, severe; red, moderate; pink, mild; white, not sensitive; and light blue, resistant. (**B**) Proportion of the 91 MER gene mutants classified into indicated groups of drug-sensitivity towards 5-FU.

**Figure 2 ijms-24-10687-f002:**
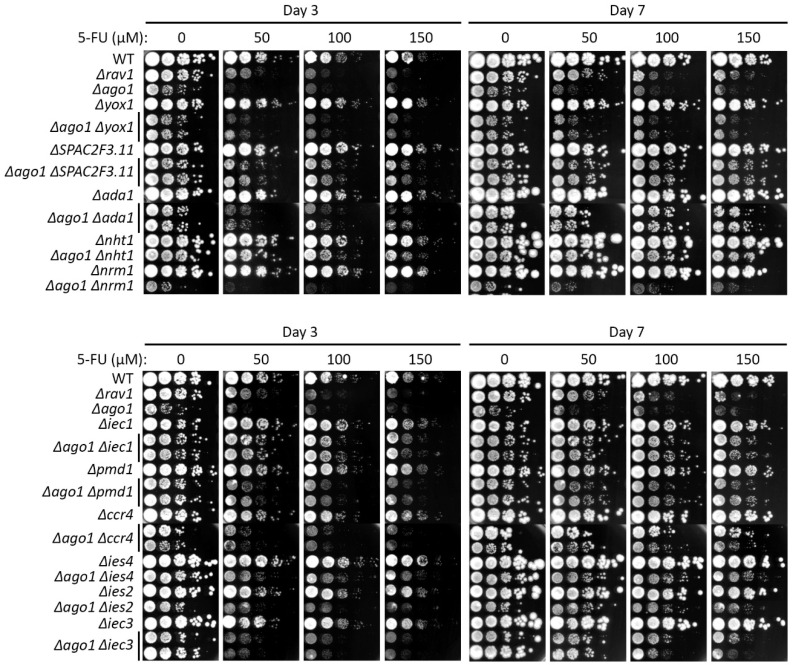
Genetic interactions of Ago1 with 5-FU-resistant MER gene mutants were tested on 5-FU. 5-FU hypersensitivity of the MER single and double mutants in combination with *Δago1* were tested at 0 (untreated), 50, 100 and 150 µM 5-FU and documented on days 3 and 7 upon drug exposure. Also refer to Appendix A for the interpretation of the hypersensitivity phenotypes.

**Figure 3 ijms-24-10687-f003:**
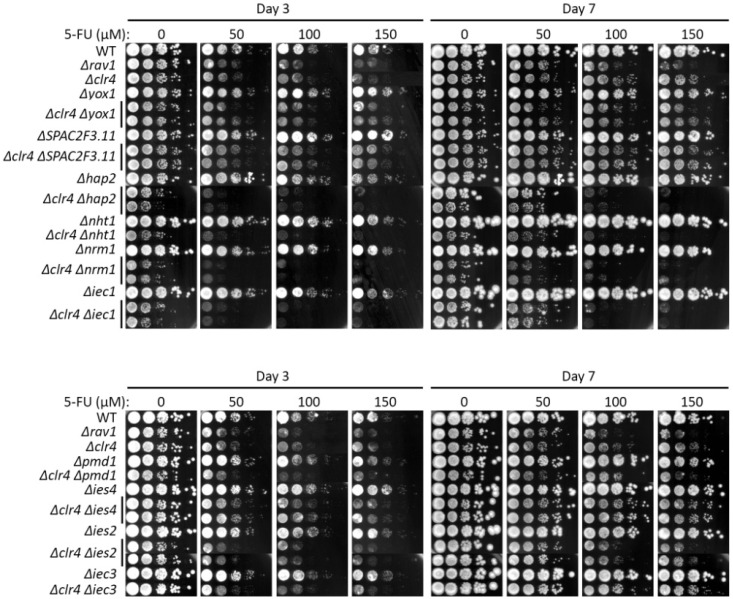
Genetic interaction of Clr4 with 5-FU-resistant MER gene mutants were tested on 5-FU. 5-FU hypersensitivity of the MER single and double mutants in combination with *Δclr4* were tested at 0 (untreated), 50, 100 and 150 µM 5-FU and documented on days 3 and 7 upon drug exposure. Also refer to Appendix A for the interpretation of the hypersensitivity phenotypes.

**Figure 4 ijms-24-10687-f004:**
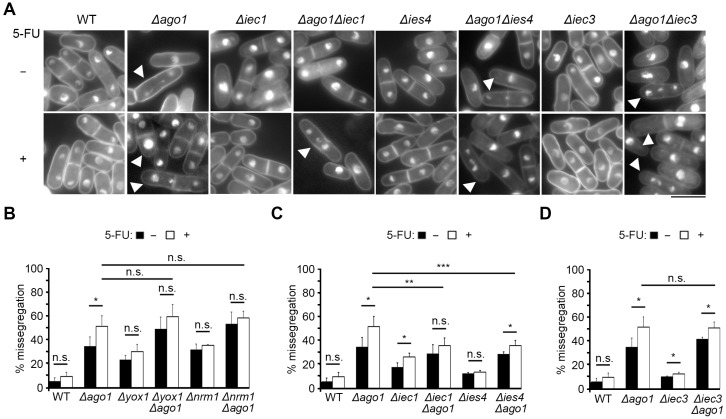
Chromosome missegregation of *Δago1* is rescued by deletion of Ino80 subunits *iec1* and *ies4*. (**A**) Nuclear morphologies of WT, *Δago1*, *Δiec1*, *Δiec3* and *Δies4* single mutants and *Δago1Δiec1*, *Δago1Δiec3*, *Δago1Δies4* double mutants were observed via staining with DAPI. Scale bar: 10 µm. −: untreated, +: treated with 500 µM 5-FU for 4 h. White arrowheads: cells showing chromosome missegregation. (**B**–**D**) proportion of cells exhibiting chromosome missegregation in single and double mutants of *Δago1* with (**B**) *Δyox1* and *Δnrm1*, (**C**) *Δiec1* and *Δies4*, and (**D**) *Δiec3*. Black bars: untreated; white bars: 5-FU treated. Bar plot indicates the mean of three experimental replicates. N > 200. Two tailed *t*-test, bar: S.D. * *p* < 0.05, ** *p* < 0.01, *** *p* < 0.001; n.s., not significant.

**Figure 5 ijms-24-10687-f005:**
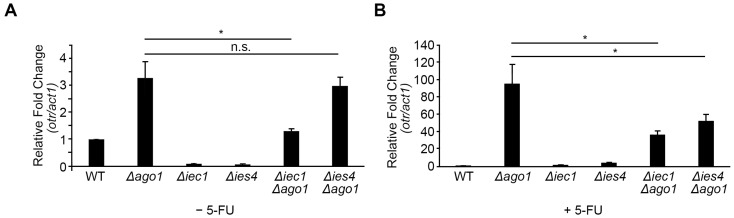
Restoration of transcriptional repression at outer centromeric (*otr*) heterochromatin of *Δago1* by *Δiec1* and *Δies4* in cells (**A**) untreated and (**B**) treated with 500 µM 4 h. Relative fold change was determined by normalizing levels of transcript from *dh* sequence in *otr* with actin gene (*act1*) detected using RT-qPCR. Bar plot indicates the mean of three experimental replicates. Two tailed *t*-test, bar: S.D. * *p* < 0.05; n.s., not significant.

## Data Availability

Not applicable.

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
