# Peer review of "Resistance to Chemotherapeutic 5-Fluorouracil Conferred by Modulation of Heterochromatic Integrity through Ino80 Function in Fission Yeast"

_ijms, 2023, doi:10.3390/ijms241310687_

Round 1

Reviewer 1 Report

In the manuscript entitled “Resistance to chemotherapeutic 5-fluorouracil conferred by modulation of heterochromatic integrity through Ino80 function in fission yeast,” submitted by Lim et al., the authors searched for the genes that conferred sensitivity and resistance to 5-fluorouracil (5-FU) in the fission yeast, Schizosaccharomyces pombe. 5-FU is a fluorinated pyrimidine analogue, which has been used in cancer therapy for a long time. Once 5-FU is converted to its metabolites in the cells, it interferes with RNA transcription, DNA replication, and maintenance of the amount of nucleotides. However, it seems that genetic determinants underlying 5-FU responsiveness are not fully understood. To understand this problem, the authors assessed proliferation of mutant fission yeast cells deleted of the multi-environmental factor responsiveness (MER) genes that were previously identified by the authors. Out of 91 MER gene-null mutants, the authors found that 12 mutants exhibited resistance to 5-FU, and that 48 mutants showed increased sensitivity. These ‘hit’ genes have a broad range of function: membrane transport, chromosome segregation, mitochondrial respiration, and chromatin remodeling. Among these hit genes, the authors focused on the role of the Ino80 chromatin remodeling complex, mutants of which showed 5-FU resistance, in maintenance of heterochromatin integrity that requires the RNA interference factor Ago1. Hypersensitivity of ago1 deletion mutant to 5-FU was rescued by deletion of iec1 or ies4 genes that encode an accessory subunit and a conserved subunit of the Ino80 complex, respectively. This rescue is attributed to partial restoration of heterochromatin silencing; while ago1 mutant showed transcription at the pericentromeric heterochromatin region otr, the transcriptional silencing at the otr was partially restored in ago1 iec1, and ago1 ies4 double mutants. The authors conclude that maintenance of chromatin structure at the pericentromere region is a mechanism to confer resistance to 5-FU.

Although several studies have been made to understand the genetic background of 5-FU action, the Ino80 complex was not identified. In this sense, the submitted manuscript reports the results of considerable interest and potential importance toward clinical application. Therefore, this manuscript is, I believe, well suitable for publication in the IJMS, after the authors respond to the points and/or questions listed below.

(1) (L68-71) As RNAi and heretochromatin are the core topics in the Results, the authors should describe more (e.g. one full paragraph) about these topics in the Introduction so that the readers can better understand the results.

(2) (L72-74) The authors should describe more (e.g. the number and the gene function) about the MER genes.  

(3) (Section 2.1) It is unclear or not explicitly described how many mutants were tested/screened. Because at the first look, 91 MER genes looked a part of the whole screening, insert a sentence describing the number of the mutants tested/screened. Also, add a sentence explaining why the whole genome screening was not performed in this study.

(4) (Figure 1A) If I understand this study correctly, Fig 1A shows all genes tested. The 91 genes are grouped by their annotated functions (e.g. Chromatin remodeling), which guides the readers nicely. However, information about detailed gene functions and complex formation is missing, though it is important to think about the phenotype of 5-FU sensitivity/resistance. Addition of a supplementary table showing the screening S-score, the group name, gene description (e.g. from pombase), and complex formation (e.g. from pombase) for all the genes tested would be necessary for the readers to understand the results.

(5) (L125) Reconsider this wording “low hypersensitivity.”

(6) (Figures 2 and 3) Because some mutants showed poor proliferation in the absence of 5-FU (0 uM), interpretation of the figure is complicated. To help the readers understand the result, the authors should add some marks showing the extent of proliferation (e.g. -, +/-, +, ++, +++) next to the figure panels.

(7) (Figures 2 and 3) While mutants used in these figures are described in the Results, many readers would not be able to understand the identity of the mutants used. I strongly recommend to prepare a supplementary list of the genes and their details (e.g. gene function and complex formation, from pombase).

(8) (L420-421) Good discussion. I hope that the 5-FU resistance could be overcome in the future.

Minor points.

(9) (L79) resistance -> resistant

(10) (L110) non-sensitive -> non-sensitive/resistant

(11) (L131) fission yeast, as in human (add a reference here).

(12) (L148) This have not been noticed -> As this has not been noticed

(13) (L306) may underlies -> may underlie. 

I do not have specific comments on the quality of English Language for the submitted manuscript. For some minor typos or missing words, please refer to the minor points of my comments to the authors. 

Author Response

(1) (L68-71) As RNAi and heretochromatin are the core topics in the Results, the authors should describe more (e.g. one full paragraph) about these topics in the Introduction so that the readers can better understand the results.

Response: We have included a paragraph on RNAi and heterochromatin in the introduction section as suggested.

(2) (L72-74) The authors should describe more (e.g. the number and the gene function) about the MER genes. 

Response: We have included a few more lines in the last paragraph of the introduction to explain about the MER genes. The functions of individual MER genes were also included in Supplementary Table 1.

(3) (Section 2.1) It is unclear or not explicitly described how many mutants were tested/screened. Because at the first look, 91 MER genes looked a part of the whole screening, insert a sentence describing the number of the mutants tested/screened. Also, add a sentence explaining why the whole genome screening was not performed in this study.

Response: We have included more information in the last paragraph of the introduction to explain about our intention of using the small collection of MER gene mutants for our current screen. The reviewer is correct that we only screened 91 mutants. The current screening is actually a pilot test. Because of our approach of wanting to use manual serial dilution spot test, which we have found to be the most reliable approach to analyze the drug hypersensitivity phenotypes of mutants, and due to the extent of work involved, we only focused on the 91 MER gene mutants in this pilot test. These MER mutants were chosen because we have done extensive test and ensure their genetic background and gene disruption. Our previous work revealed that many mutants in the whole genome null mutant collection were actually not as annotated as many are not properly disrupted. Furthermore, the multi-nutritional marker background can greatly confound on the phenotypes (information regarding confounding genetic background have been published in Tay et al, PLoS ONE, 2013). Because of the unreliability of the mutant library, we believe that previously reported screens contain many false negative data. But because we also did not test every strain in the library, we cannot mention the unreliability of the library in our text, even though this is a known fact in the research field.

(4) (Figure 1A) If I understand this study correctly, Fig 1A shows all genes tested. The 91 genes are grouped by their annotated functions (e.g. Chromatin remodeling), which guides the readers nicely. However, information about detailed gene functions and complex formation is missing, though it is important to think about the phenotype of 5-FU sensitivity/resistance. Addition of a supplementary table showing the screening S-score, the group name, gene description (e.g. from pombase), and complex formation (e.g. from pombase) for all the genes tested would be necessary for the readers to understand the results.

Response: We have included this table as new Supplementary Table 1 in the revised manuscript as suggested. 

(5) (L125) Reconsider this wording “low hypersensitivity.”

Response: We have amended to “low sensitivity”. 

(6) (Figures 2 and 3) Because some mutants showed poor proliferation in the absence of 5-FU (0 uM), interpretation of the figure is complicated. To help the readers understand the result, the authors should add some marks showing the extent of proliferation (e.g. -, +/-, +, ++, +++) next to the figure panels.

Response: Because these figures contain multiple strips of spotting results at different concentrations, we think it will complicate the figure if we adding the + and – symbols beside each spotting. Instead, we have included these symbols in the new Supplementary Tables 2 and 3.  

(7) (Figures 2 and 3) While mutants used in these figures are described in the Results, many readers would not be able to understand the identity of the mutants used. I strongly recommend to prepare a supplementary list of the genes and their details (e.g. gene function and complex formation, from pombase).

Response: We have included this table as new Supplementary Table 1 in the revised manuscript as suggested. 

(8) (L420-421) Good discussion. I hope that the 5-FU resistance could be overcome in the future.

Response: We thank the Reviewer for the encouraging comment.  

Minor points.

(9) (L79) resistance -> resistant

Response: We would like to retain ‘resistance’. 

(10) (L110) non-sensitive -> non-sensitive/resistant

Response: We have made the amendment.  

(11) (L131) fission yeast, as in human (add a reference here).

 Response: We have included the references. 

(12) (L148) This have not been noticed -> As this has not been noticed

  Response: We have made the amendment. 

(13) (L306) may underlies -> may underlie. 

  Response: We have made the amendment. 

Comments on the Quality of English Language

I do not have specific comments on the quality of English Language for the submitted manuscript. For some minor typos or missing words, please refer to the minor points of my comments to the authors. 

Submission Date

25 May 2023

Date of this review

14 Jun 2023 03:49:15

Reviewer 2 Report

This work has conducted a focused genetic study in fission yeast, exploring the mechanisms of action for 5-FU, a traditional drug for cancer chemotherapy. Starting with a panel of mutants – multiple environmental factors (MEF), the authors have identified several categories of genes, the mutations of which exhibit hypersensitivities; or, in contrast, resistance to 5-FU. Further studies of genetic interaction (suppression) among the identified mutations and cellular phenotype characterization have led authors to conclude that compromising the chromosome segregation fidelity via perturbing the centromeric chromatin structure is likely a novel mechanism through which 5-FU causes cytotoxicity.

One outstanding strength of this study is the identification of clustered mutations (mutations of genes of the same protein complex, or, in the same genetic pathways), whose phenotypes validate and reinforce each other. In the case of 5-FU resistant mutants, individually, the resistance phenotype may appear debatable, but collectively, especially in conjunction with suppression of ago1- hypersensitivity, it is convincing that the ino80 complex mutations are resistant to 5-FU.

Furthermore, multiple lines of evidence are consistent with ino80 mutations conveying drug resistance by impacting pericentromeric heterochromatin. Strictly speaking, the phenotypes of transcription silencing and TBZ sensitivity as well as chromosome mis-segregation, are in parallel with, but falling short of proving causative relationship for 5-FU sensitivity.

Overall, this work is extensive and the novel findings, although preliminary in molecular mechanisms, point to new directions for future investigations of 5-FU cytotoxicity. I am supportive for its publication, provided that the following specific points are addressed.

1.     A brief introduction (and the reference) of “MEF” mutants and the rationale of focusing on them for 5-FU susceptibility study should be provided. Furthermore, multiple studies in fission yeast have reported finding mutants that are sensitive to 5-FU (but no report for 5-FU resistant mutant??). A summary of these studies and a highlight of the new experimental approach for this study in Introduction would be helpful.

2.     Fig. 1A figure label should be “Day 7” instead of “Day 5”?

3.     Lines 223/224, “once varied” should be “once again varied”?

4.     Lines 224 – 229, these sentences are confusing and hard to follow.

5.     Lines 253/254, this sentence is also confusing – “comparable” refers to …?

6.     Some of the remarks should be toned down, for example, Line 375 “prominent” drug resistance; in particular, the final conclusion at Lines 418/419 is an overstatement.

7.     Deletion of nrm1 and yox1 has been shown to cause up-regulation of cnp1/CENP-A transcription. Authors might want to consider a possible close connection with the current working model.

please see above

Author Response

  1. A brief introduction (and the reference) of “MEF” mutants and the rationale of focusing on them for 5-FU susceptibility study should be provided. Furthermore, multiple studies in fission yeast have reported finding mutants that are sensitive to 5-FU (but no report for 5-FU resistant mutant??). A summary of these studies and a highlight of the new experimental approach for this study in Introduction would be helpful.

Response: Thank you for the suggestion. We have included more information in the last paragraph of the introduction section regarding the MER mutants and an explanation of why we wanted to focus only on screening the smaller collection of MER mutants (as suggested by Reviewer 1). Because of the flow of the argument, we find it difficult to include the summary of previously reported screens in the introduction section but instead we devote a significant portion of the Discussion section to these studies. Having said so, we have added information that earlier screen has been reported by Morjadin et al, 2015 in the last paragraph of the introduction and mention that many of the 5-FU hypersensitivity mutants we identified have also been identified by them. This information is again mentioned in the Results section.

  1. Fig. 1A figure label should be “Day 7” instead of “Day 5”?

Response: We have amended the mistake.

  1. Lines 223/224, “once varied” should be “once again varied”?

Response: We have made the amendment.

  1. Lines 224 – 229, these sentences are confusing and hard to follow.

Response: We have made some amendment to the section and hopefully it is now easier to follow.

  1. Lines 253/254, this sentence is also confusing – “comparable” refers to …?

Response: “Comparable” refers to similar level of chromosome missegregation being observed between single Δago1 mutant compared to the Δyox1Δago1 and Δnrm1Δago1 double mutants. We have made some amendment to the description, which hopefully clarify the relationship.

  1. Some of the remarks should be toned down, for example, Line 375 “prominent” drug resistance; in particular, the final conclusion at Lines 418/419 is an overstatement.

Response: We have deleted all “prominent” words and the lines 418/419 from the text.

  1. Deletion of nrm1 and yox1 has been shown to cause up-regulation of cnp1/CENP-A transcription. Authors might want to consider a possible close connection with the current working model.

Response: In our results Figure 4A, deletion of nrm1 and yox1 did not affect the phenotype of Δago1 and neither show a dependency on the presence of 5-FU, making us think that the up-regulation of cnp1+ transcription in Δyox1 and Δnrm1 may not have a close connection to the modulation of 5-FU response by Ago1 and Ino80. Furthermore, current view positions heterochromatin factors including RNAi factors to act upstream to establish CENP-A onto nascent inner core centromeric sequences (Folco et al, Science, 2008) and there is little evidence that CENP-A affects heterochromatin assembly at the pericentromeric regions and other constitutive heterochromatic loci. Based on this knowledge, we believe a change in cnp1+ transcript is also unlikely to affect Ago1 function at the heterochromatin, leading us to think that there is no close connection between the cnp1/CENP-A transcription and our working model at this stage.